# The Secondary Myelodysplastic Neoplasms (MDS) Jigsaw

**DOI:** 10.3390/cancers15051483

**Published:** 2023-02-26

**Authors:** Oriol Calvete, Julia Mestre, Andrés Jerez, Francesc Solé

**Affiliations:** 1MDS Group, Institut de Recerca Contra la Leucèmia Josep Carreras, 08916 Barcelona, Spain; 2Experimental Hematology Unit, Department of Hematology, Vall d’Hebron Institute of Oncology (VHIO), University Hospital Vall d’Hebron, 08035 Barcelona, Spain

**Keywords:** myelodysplastic neoplasms (MDSs), secondary MDS, genetic predisposition, therapy-related myeloid neoplasm (TRMN), MDS post-cytotoxic therapy (MDS-pCT), MDS comorbidity, clonal hematopoiesis of indeterminate potential (CHIP)

## Abstract

**Simple Summary:**

The classification of myelodysplastic neoplasms (MDSs) in patients with a previous primary tumor is controversial, as an efficient consensus risk factor-based classification of secondary MDSs has not yet been established. The current classifications consider separate etiologies based on exposure to cytotoxic therapy and genetic predisposition, despite recent studies on the germline landscape and clonal hematopoiesis of indeterminate potential (CHIP) supporting the contention that these different risk factors are overlapping. This review wants to summarize the current status of knowledge of secondary MDS etiologies and foresees future classifications that assess all risk factors and their interactions for achieving a differential diagnosis of patients at risk to aid in routine clinical decision-making related to most adequate clinical management.

**Abstract:**

There is a great deal of controversy in the hematologic community regarding the classification of secondary myelodysplastic neoplasms (MDSs). Current classifications are based on the presence of genetic predisposition and MDS post-cytotoxic therapy (MDS-pCT) etiologies. However, since these risk factors are not exclusive for secondary MDSs and there are multiple overlapping scenarios, a comprehensive and definitive classification is yet to come. In addition, a sporadic MDS might arise after a primary tumor fulfills the diagnostic criteria of MDS-pCT without a causative cytotoxicity. In this review, we describe the triggering pieces of a secondary MDS jigsaw: previous cytotoxic therapy, germline predisposition and clonal hematopoiesis. Epidemiological and translational efforts are needed to put these pieces together and ascertain the real weight of each of these pieces in each MDS patient. Future classifications must contribute to understanding the role of secondary MDS jigsaw pieces in different concomitant or independent clinical scenarios associated with the primary tumor.

## 1. Introduction

Myelodysplastic syndromes or, as renamed in the latest World Health Organization (WHO) classification [1], myelodysplastic neoplasms (MDSs) are an age-associated malignant condition characterized by ineffective hematopoiesis, which entails a variety of prognostic biomarkers and heterogeneous outcomes due to their diverse etiology and the risk factors involved [2]. However, the landscape of the risk factors driving different MDS etiologies is not fully understood. Previous studies reported both obesity and smoking as modifiable risk factors of MDSs [3]. Furthermore, exposure to cytotoxic therapies [4], genetic predisposition [5] or the presence of clonal hematopoiesis of indeterminate potential (CHIP) [6] have been described as risk factors for this pathology.

The development of a secondary myelodysplasia is benign and often reversible if the offending trigger is withdrawn [7]. In Vienna, in 2006, the Myelodysplastic Neoplasm Working Group conference consolidated the two prerequisite criteria for the diagnosis of MDSs: the presence of cytopenia and the absence of other hematopoietic or nonhematopoietic disorders as the etiology of cytopenia [8]. Further contributing to the confusion, the term secondary myelodysplasia was also being used to refer to MDS evolving secondarily from a previous myeloid neoplasm or to define diseases that progressed from another myeloid disease. However, in the last WHO classification, the transformation of a myeloproliferative neoplasm (MPN) to acute myeloid leukemia (AML) was retained under the MPN category, while the transformation of MDS to AML and MDS/MPN remains under AML-myelodysplasia-related (AML-MR), in the secondary myeloid neoplasm category [1]. In very early reports, Jens Pedersen-Bjergaard [9] first described secondary myeloid neoplasms in the study on “Acute Nonlymphocytic Leukemia, Preleukemia, and Acute Myeloproliferative Syndrome Secondary to Treatment of Other Malignant Diseases”, in which among 31 patients, 21 “pre-leukemia” cases predominated. The term “secondary MDS” was first used in “Proposals for the classification of the myelodysplastic syndromes”, in which the FAB group addressed the “Special features of Secondary MDS”, describing a more frequent presence of fibrosis, hypocellularity, ringed sideroblasts, a higher proportion of blasts in the peripheral blood (PB) than would be expected from the percentage in the bone marrow (BM) and abnormal and immature megakaryocyte precursors often seen in PB and BM [10]. Using sequential cytogenetics, Rowley and colleagues demonstrated in 1980 that the development of MDS after exposure to mutagens and carcinogens was related to chromosomal evolution towards complex karyotypes (–5/del(5q), –7, +8, +21) and cytological transition into acute leukemia [11].

## 2. The Pieces of the Jigsaw

### 2.1. MDS Post-Cytotoxic Therapy

MDS post-cytotoxic therapy (MDS-pCT) cases are aggressive hematologic malignant neoplasms. The incidence of these neoplasms is rare (<0.5 per 100,000), but the mortality rates are higher compared to primary MDS, with a 5-year survival rate of 10% vs. 31% in primary MDS, and a median survival of around 8–10 months [12]. The diagnosis of MDS-pCT requires fulfilment of the criteria for MDS in addition to a previous history of chemotherapy treatment or large-field radiation therapy for an unrelated neoplasm [4]. Both incidental and therapeutic radiation have been associated with MDS-pCT. Indeed, a significant linear radiation dose-response for MDS has been described in atomic bomb survivors 40 to 60 years after radiation exposure [13]. In the cancer setting, a study of patients receiving radiation therapy showed a higher risk of developing MDS than in those who did not [14]. Nevertheless, the conclusions of studies in specific cancer series are not very uniform, with conflicting data in relation to breast cancer, Hodgkin lymphoma and radiation as part of myeloablative regimens before hematopoietic stem cell transplantation [15].

It has been estimated that 10% of patients with non-Hodgkin lymphoma [16], 8.2% of patients with chronic lymphocytic leukemia [17] and 3.4% of patients with multiple myeloma (MM) develop MDS-pCT [18]. Cytogenetic abnormalities are detectable in almost 90% of MDS-pCT patients [19], while altered karyotypes are reported in 40–50% of patients with de novo MDS [20]. Likewise, while high-risk forms with a poor prognosis are prevalent in the 46–70% of MDS-pCT patients [20,21], they are only reported in 30% of patients with de novo MDS [22]. However, the same cytogenetic abnormalities are found in both groups of patients, who were undistinguishable at the karyotype level [19,23,24] (Table 1).

On the other hand, the latency period for the appearance of MDS-pCT after treatment substantially varies depending on the type of primary cancer and the treatment regimen [21]. This is well known for drugs used for long periods in anticancer schedules. Treatment with alkylating agents has been associated with longer latency times, adverse cytogenetics with a high frequency of complex karyotypes and a poor prognosis [28]. Regarding cytogenetics, the most common clonal abnormalities include the partial or total loss of chromosomes 5 and 7 and complex karyotypes after treatment with alkylating agents [22]. The development of MDS-pCT after receiving anthracyclines and/or topoisomerase II inhibitors is associated with a median latency of 1 to 3 years and an *MLL* translocation at 11q23 or *RUNX1/AML1* at 21q22 and a low frequency of complex karyotypes [22,26,27] (Table 1).

Exposure to PARP1 inhibitors is another criterion for the development of MN-pCT [1]. Poly (ADP-ribose) polymerase (PARP) inhibitors (PARPi) are active in cells with impaired ability to repair DNA double-strand breaks, such as cancer cells with mutations in the tumor suppressors *BRCA1* or *BRCA2*, and have been used in subsets of patients with ovarian, breast, prostate and pancreatic cancer [28,29]. During the last 2 years, several reports have identified patients with MDS and AML following PARPi therapy [1,24,30]. Common findings include a median latency of 2 years after the initiation of PARPi treatment, complex karyotypes, the presence of germline damage response gene variants and the acquisition of *TP53* mutations [31,32]. Interestingly, one of these studies reported how clonal hematopoiesis was more common in patients with ovarian cancer receiving PARPi maintenance therapy than in those not receiving this treatment, showing expansion in paired specimens pre- and post-therapy [33].

On the other hand, immunomodulatory drugs were introduced for treating MM in the late 90s, leading to significantly improved overall survival. During the following decade, lenalidomide replaced thalidomide as the most used immunomodulatory drug for MM due to its higher efficacy and lower toxicity. Several clinical trials have found significantly higher rates of secondary myeloid neoplasms in lenalidomide-treated arms than in those without lenalidomide in relapsed/refractory, transplant-eligible and transplant ineligible MM patients [34]. A definitive role for lenalidomide in the development of a secondary myeloid neoplasm after therapy for MM cannot be established in many cases since patients often receive high-dose chemotherapy during their initial treatment. However, a recent systematic analysis of 416 patients with MN-pCT and a detailed prior history of exposure found that *TP53* mutations were significantly associated with previous treatment with thalidomide analogs, specifically lenalidomide. They also showed that lenalidomide treatment provides a selective advantage over *TP53* (Trp53 in mice)-mutant murine hematopoietic stem and progenitor cells (HSPCs) in vitro and in vivo, and that the effect was specific to Trp53-mutant HSPCs and was not observed in HSPCs with other clonal hematopoiesis mutations [35]. A recent systematic review and meta-analysis found that lenalidomide-induced later sporadic malignancies seem to occur exclusively in patients with MM, and no significant increase was described in chronic lymphocytic leukemia and MDS trials [36].

Scarce information has been published regarding prognostic models or associations between MDS-pCT and concrete clinical outcomes and progression [37]. In addition, there are no clinical management guidelines for the appropriate monitoring of these patients, since there are no accurate diagnostic criteria that consider all etiologic factors involved in the development of MDS-pCT, thereby hampering the exploration of targeted therapies. An important clue when identifying the different etiologies in secondary MDS relies on the fact that the somatic signatures in MDS-pCT are indistinguishable from those occurring in de novo MDS [12], which would hide the incidence of the treatment effect in MDS development [38]. Only *TP53* mutations were found to be enriched in MDS-pCT patients, while spliceosome mutations are more frequent in de novo MDS, which might partially explain the complex karyotype and unfavourable clinical outcomes of MDS-pCT patients [39]. Likewise, mutations in the *PPM1D* gene, a negative regulator of the DNA damage response pathway, are also frequent in MDS-pCT patients, as they confer a competitive advantage under the selective pressure of chemotherapy [40].

### 2.2. Germline Predisposition

Over the last decade, large-scale genomic studies have described the landscape of genomic variants in many of the most relevant types of cancer with the initial and fundamental objective of providing prognostic, diagnostic and pathogenic information based on the acquired alterations detected. However, the co-assessment of germ tissue in these series has transformed the understanding of how inherited variants influence cancer development [41]. Within myeloid neoplasms, it was estimated that 5% to 10% of patients with AML carried germline variants predisposing them to myeloid neoplasia [42]. In MDS, it has been estimated that germline mutations could explain at least 15% of adult and pediatric MDS cases [43]. In specific contexts, such as adolescents with MDS and monosomy of chromosome 7, this percentage could reach up to 70% [44].

The identification of clinical features and molecular biomarkers linked to this entity is important since its clinical management differs from sporadic MDS [21]. To this end, several guidelines for myeloid neoplasms with a germline predisposition have been described to identify these patients [45,46]. A growing number of inherited genetic loci that contribute to MDS has been identified [47,48]. The *SAMD9*, *SAMD9L*, *SRP72*, *TERC* and *TERT* genes, together with other genes typically mutated in sporadic MDS, such as *TP53*, *GATA2*, *DDX41*, *ANKRD26*, *ETV6*, *CEBPA*, *ASXL1* and *RUNX1*, have been associated with the germline development of the disease [49]. Different correlations have been established among germline mutations, the age of onset and the severity of the myeloid neoplasm [50].

Furthermore, several hereditary syndromes have also been associated with MDS development [51]. These MDSs arise within the genetic landscape that predisposes patients to multiple tumors [52]. Several diseases, such as Fanconi Anemia, Severe Congenital Neutropenia, Dyskeratosis Congenita or Blackfan–Diamond Anemia present with bone marrow failure triggering MDS [53]. Although a particular karyotype has not been described in most germline-predisposed MDSs, in the context of inherited bone marrow failure (BMF) syndromes, recurrent chromosomal findings have been described, including duplications of chromosome regions 1q or 3q in Fanconi Anemia, isochromosome 7q in Shwachman–Diamond Syndrome, or isolated monosomy 7 common in *GATA2* haploinsufficiency, among others [54] (Table 1).

### 2.3. Clonal Hematopoiesis

Finally, the presence of clonal hematopoiesis has been observed to increase with age and actively participates in the development of myeloid neoplasms [55]. Clonal hematopoiesis of indeterminate potential (CHIP) is defined as the presence of clonal mutations in genes recurrently mutated in myeloid neoplasms in peripheral blood of healthy individuals at a low frequency [6,56]. The incidence of CHIP has been associated with a higher risk of developing hematologic malignant neoplasms with adverse outcomes [57]. Altered clones not only harbor genetic alterations but also numerical and structural chromosomal changes, including those found in hematopoietic malignancies, such as del(20q), del(13q), del(11q), trisomy 8 or less commonly, del(5q) or del(7q) [6,21] (Table 1). CHIP has recently been described as a risk factor for developing secondary cardiovascular diseases [58].

### 2.4. Current Classifications and Secondary MDS

In light of the different risk factors entailing different outcomes, the identification of risk factors must be concise for differential diagnosis and adequate clinical management [59]. However, identification of different overlapping factors in relation with the development of MDS in a patient with a previous primary tumor is challenging. This hematologic condition is considered a secondary MDS, but an efficient consensus risk factor-based classification has yet to be established.

The WHO first classified MDS occurring following cytotoxic therapy for a primary tumor as MDS-pCT independently of MDS associated with a germline predisposition [60]. The latest update of the WHO classification (2022) proposes considering the entity of secondary myeloid neoplasms, which encompass the MDSs that arise from previous exposition to cytotoxic therapy or immune intervention (MDS-pCT), as well as MDS that occurs within the context of a syndromic germline [1]. These separate subentities do not consider other risk factors or the overlapping of both conditions (MDS-pCT and germline predisposition).

On the other hand, the recent International Consensus Classification (2022) maintains the therapy-related myeloid neoplasm (TRMN) category as an entity [61]. Nevertheless, this update clarifies that TRMN should be subclassified according to its morphology and genetics, as risk factors, such as CHIP or clonal cytopenia, can occur after exposure to cytotoxic treatment. In addition, this proposal also suggests that the presence of an underlying germline condition must be explored, considering a possible relocation to germline mutation-associated disorders either as syndromes when the genetic origin is common between the two tumors or as having different molecular drivers.

Both classifications are based on fitting, including two delimited etiologies for secondary MDS, based on therapy toxicity and genetic predisposition (Figure 1A). However, this two independent etiology-based classification does not consider the genetic predisposition aside from syndromic MDSs, excluding multiple overlapping etiologies and scenarios that share the same clinical appearance. The relative contribution of risk factors, such as germline predisposition or the presence of CHIP, in the development of MDS-pCT have not yet been fully explored.

Thus, genetic predisposition might increase the susceptibility to cytotoxic agents. The treatment of a primary tumor may affect bone marrow cells differently depending on the presence of mutations affecting the DNA damage repair system. In this sense, it has been described that between 16 and 21% of cancer survivors who developed MDS-pCT had a germline mutation associated with inherited cancer susceptibility genes [12]. Several studies have reported germline mutations in *BRCA1, BRCA2*, *PALB2*, *CHEK2* and *TP53* [62,63] and Fanconi Anemia genes in MDS-pCT patients [64].

Similarly, recent studies showed that cancer therapy shapes the fitness landscape of clonal hematopoiesis by promoting the onset or the increment of cytopenia and clonal dysplasia, such as idiopathic cytopenia of undetermined significance (ICUS), clonal cytopenia of undetermined significance (CCUS), idiopathic dysplasia of unknown significance (IDUS) or CHIP [65]. CHIP was higher than expected according to age in patients with MDS-pCT at the time of diagnosis of the primary tumor and before treatment [66]. According to recent studies, it is estimated that 30% of patients with MDS-pCT have CHIP [65]. CHIP was detected in 66% of MDS-pCT patients previously treated for gynecologic and breast cancers, including mutations in *TP53* (31%), *DNMT3A* (19%), *IDH1/2* (13%), *NRAS* (13%), *TET2* (12%), *NPM1* (10%), *PPM1D* (9%) and *PTPN11* (9%) [39]. Since CHIP is frequently observed in patients with MDS-pCT at the time of diagnosis of the primary tumor, it has been suggested as a predictive marker to identify patients at risk to preclude the administration of treatments that might trigger the development of MDS-pCT [67].

In addition to these overlapping scenarios, both primary and secondary tumors might not be associated with any risk factor and do not occur concomitantly. Current therapeutic strategies have reduced the mortality among cancer patients, with an increase in survival rates entailing an increased frequency of age-dependent secondary pathologies with no relationship with the primary tumor. Thus, both tumors might arise independently or from different risk factors. The MDSs of these patients would have no relationship with the cytotoxic therapy of the primary tumor or a common genetic origin (syndrome), but they mimic the clinical definition of a secondary MDS.

## 3. Discussion: How to Put the Jigsaw Pieces Together

With a few exceptions, the cytogenetic abnormalities and molecular similarities between secondary MDSs of different etiologies hamper adequate classification of differential diagnoses [20,21,36] (Table 1). Furthermore, only a few studies with low number of patients have evaluated the proportion of altered karyotypes within the etiologies of CHIP and genetic risk factors within the context of secondary MDS (Table 1). Thus, the debate regarding the classification of secondary MDS is apparently going to continue during the coming years. The current classifications consider separate etiologies based on risk factors and cytotoxic therapy to describe either secondary MDS [1] or TRMN [61] (Figure 1A). Nevertheless, recent studies regarding the germline landscape and CHIP condition in MDS-pCT patients support the contention that these different etiologies must be considered as overlapping [37,60,66,67] (Figure 1B). Thus, genetic predisposition must be considered beyond bone marrow syndromes [12]. Likewise, CHIP may be initiated, expanded or triggered by cytotoxic therapy [67]; however, both CHIP and treatment for the primary tumor could also coexist with no association. In addition, different mutational burdens and their correlation with CHIP-related mutations are different for different hematological malignancies [68]. Thus, different MDS subtypes may be expected in different combinations of cytotoxic effects together with an individual genetic landscape and CHIP mutations.

We foresee future classifications of secondary MDSs assessing all risk factors and their interactions for a better assessment of the etiology of the cancer. Therefore, overlapping risk factors (CHIP and germline predisposition), together with the cytotoxic therapy effect, would result in the clinical definition of different concomitant scenarios, including a single or variety of responsible risk factors. Finally, clinical scenarios in which both tumors are independent or do not share risk factors must be considered as they mimic the clinical manifestation of secondary MDS but must be identified to prevent background noise.

Therefore, we suggest the following scenarios (Figure 2).

**(a) Concomitant tumors** are related to secondary MDS when it involves the etiology of the primary tumor:

(a.1) MDS-pCT disease; the hematologic disease arises through exposure to a cytotoxic therapy when treating a primary tumor. Treatment might do the following:solely contribute to MDS without other known risk factors;trigger CHIP or drive an increase in CHIP;contribute depending on germline predisposition or susceptibility;contribute together with CHIP and the germline landscape.

(a.2) Syndrome; secondary MDS arises from a common genetic origin together with the first tumor.

**(b) Independent tumors** mimic the clinical appearance of the previous scenarios but do not share risk factors:Sporadic, correlative tumors with no shared risk factors;MDS-pCT-like, in which cytotoxic therapy does not participate in the development of a secondary myeloid tumor.

Secondary MDS occurs with germline predisposition, in which a primary tumor might also have a genetic predisposition but is not common with that of the MDS.

## 4. Conclusions and Future Directions

In summary, the role of risk factors in the different etiologies of MDS scenarios is still unclear. Furthermore, the classification of the etiologies of MDS becomes even more challenging when it occurs in a patient with a previous primary tumor. The first step to understand the molecular pathophysiology and the role of risk factors in different secondary MDSs is to have an adequate classification even with no prognostic biomarkers or molecular descriptions. Current classifications are based on etiologies related to well-delimited risk factors and cytotoxic therapy. However, the current knowledge regarding the biology of hematological malignancies supports the integration of genetic predisposition and CHIP into future MDS-pCT classifications to reflect the biological diversity and etiologies of MDS-pCT and their impact on outcomes. Thus, future classifications must consider the concomitance of single and overlapping risk factors, as well as independence regarding the relationship between primary and secondary tumors.

Deciphering the contribution of risk factors in combination with cytotoxic treatments will allow for the differential diagnoses of patients at risk to aid in routine clinical decision-making at the translational level to provide adequate clinical management. In addition, understanding the pathologic mechanisms underlying different etiologies will improve the development of prognostic biomarkers and therapy-oriented guidance of primary tumors for patients at risk of developing secondary MDS and will also help in genetic counseling related to the suitability of hematopoietic progenitor cell transplants (HPTs) from a related donor.

## Figures and Tables

**Figure 1 cancers-15-01483-f001:**
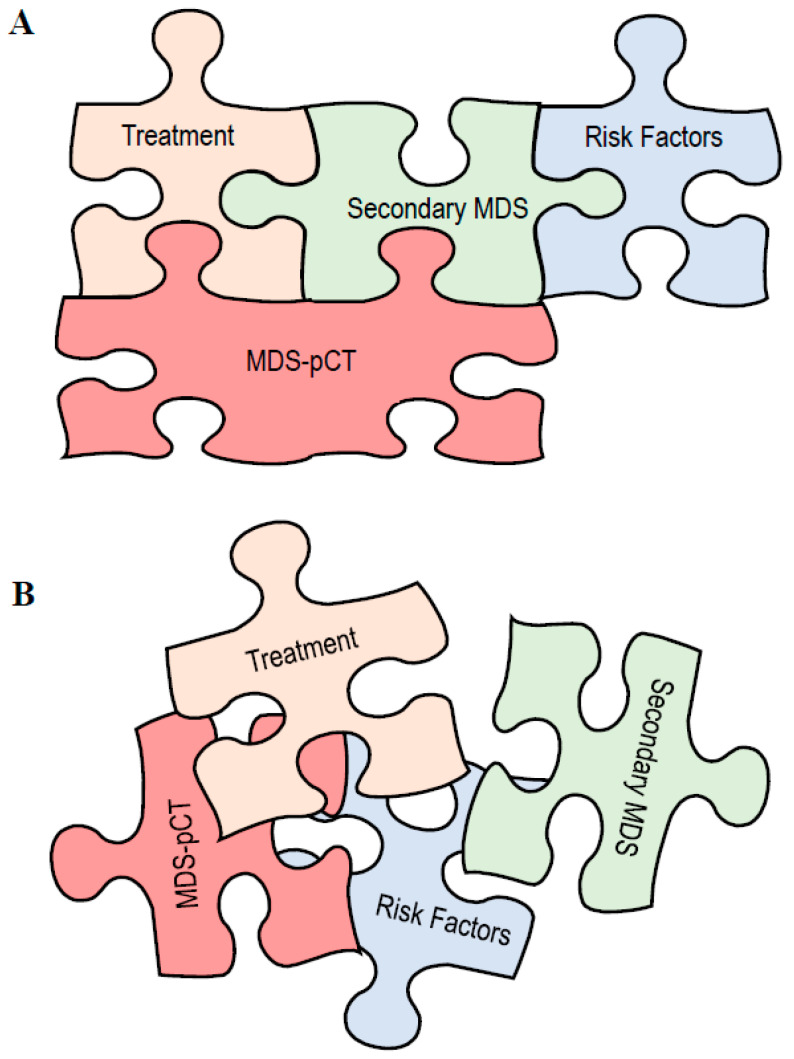
Secondary myelodysplastic neoplasm (MDS) jigsaw pieces. (**A**) Current classifications based on delimited etiologies for secondary MDS arising from cytotoxic treatment or risk factors (genetic predisposition and clonal hematopoiesis of indeterminate potential, CHIP). MDS post-cytotoxic therapy (MDS-pCT) or therapy-related myeloid neoplasm (TRMN) were secondary MDSs arising from treatment effects with no overlap with other risk factors. (**B**) New classification approach. Both cytotoxic treatment and the presence of an underlying germline or CHIP conditions are overlapping etiologies and compose different susceptibility scenarios. The different contribution of several risk factors might participate in the development of MDS-pCT. Thus, treatment might promote or increase CHIP, while genetic predisposition might modulate the effect of cytotoxic therapy.

**Figure 2 cancers-15-01483-f002:**
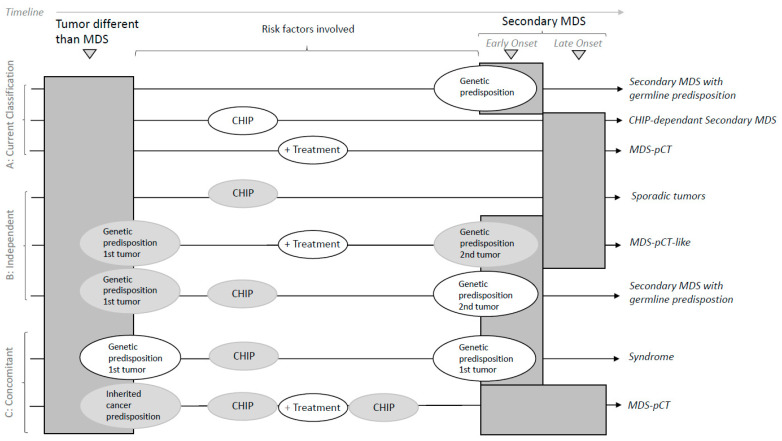
Classification of secondary MDS. The risk factors involved in different etiologies are shown for the primary tumor and secondary MDS. Risk factors in white circles refer to mandatory conditions per the scenario described. Risk factors in grey circles refer to non-strictly necessary overlapping conditions co-occurring with the mandatory condition. The timeline shows the chronology of tumor events. The expected time of onset is shown for secondary MDS. Early onset is expected when MDS involves genetic predisposition. (**A**) Current classifications are based on well-delimited risk factors (genetic predisposition, CHIP or cytotoxic treatment). New classification approaches must consider the overlap between risk factors that compose different susceptibility scenarios of independent and concomitant tumors. (**B**) Independence: both tumors are independent when they do not share risk factors (sporadic) or even when the treatment of the first tumor is not related to the development of MDS (MDS-pCT-like). In addition, the primary tumor and secondary MDS might arise from a strong but not shared genetic predisposition. (**C**) Concomitancy is considered when secondary MDS arises in relation to the etiology of the first tumor. Typically, secondary MDS with germline predisposition within a syndromic scenario arises from genetic predisposition in common with that of the first tumor. MDS-pCT arises from exposure to a cytotoxic therapy for treating the primary tumor. The relative contribution of risk factors, such as germline predisposition or the presence of CHIP, might contribute to the development of MDS-pCT. Treatment might promote or increase CHIP, while genetic predisposition might modulate the cytotoxic effect.

**Table 1 cancers-15-01483-t001:** Chromosome alterations associated with MDS with different etiologies.

Karyotype	Genetic Risk Factors	CHIP	De Novo MDS	MDS-pCT
**Altered karyotypes**	Not described	21% [25]	40–60% [20]	70–90% [19,20]
**Complex karyotypes ***	Not described	Not described	30% [22]	46–70% [12,21]
**Most frequent unique alterations**	dup(1q), 3q+, −7/del(7q), i(17)(q10), +8, +21, del(20q), del(11q) [24]	del(20q), del(13q), del(11q), +8, del(5q), del(17p) [6,12]	del(5q), −7/del(7q), +8, −Y [20,23]	Post-alkylating agents: −7, del(7q), del(5q), −5 [22,26] Topoisomerase II inhibidors: t(11;21)(q23;q22), t(15;17), inv(16)(p13q22), t(17;19)(q22;q12) [24]
**Most frequent complex karyotypes**	Not described	Not described	del(5q), −7/del(7q), −18/−18q (7%), +8, −20q. Other: +1/+1q, −5, +11, −13/13q−, −17/17p−, −21, +mar [27]	−5/del(5q), −7/del(7q) Other: der(21q), +8, der(12q), t(1;7), −12, der(17q), der(3q), der(3q), and −18 [22]

* Three or more cytogenetic alterations. MDS: myelodysplastic neoplasm; CHIP: clonal hematopoiesis of indeterminate potential; MDS-pCT: MDS post-cytotoxic therapy.

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
