# Peer review of "The Secondary Myelodysplastic Neoplasms (MDS) Jigsaw"

_cancers, 2023, doi:10.3390/cancers15051483_

Round 1

Reviewer 1 Report

The manuscript “The Secondary Myelodysplastic Neoplasms (MDS) Jigsaw” by

Calvete et al is a superficial review of secondary MDS in the context of a new classification approach in which cytotoxic treatment administration and underlying germline predisposition or clonal hematopoiesis of indeterminate potential (CHIP) contribute to secondary MDS. The manuscript needs some editing by a native English speaker to improve its readability. Moreover, the authors do not analyze several of the abbreviations used at first appearances, such as ICUS (idiopathic cytopenia of undetermined significance), CCUS (clonal cytopenia of undetermined significance), and IDUS (idiopathic dysplasia of unknown significance). The figures are not particularly helpful, especially Figure 2, which is confusing despite a decent legend, i.e., text accompanying the figure. I would also expect a greater in-depth analysis of the pathophysiology of MDS-pCT.

Author Response

We appreciate the comments and suggestions of reviewer 1. We would like to highlight that the present manuscript intends to review the current problem regarding the classification of the different etiologies driving secondary MDS. We are not deep describing the full pathophysiology or clinics but including a proposal for how to solve the different and overlapping scenarios. However, as the reviewer 1 suggests, we have included a brief sentence regarding the particular pathophysiology and clinical management of the MDS-pCT patients (see text in red in pages 2 and 4). We also agree with the other specific comments of the reviewer, we have checked the abbreviations used at first appearance through the whole text and modified accordingly. We have also revised the legend of the Figure 2 to clarify the explanation of the different scenarios (see text in red in Figure 2 Legend). In addition, we have tried to clarify the Figure by adding a “Risk Factors Involved” heading. Wee have uploaded a revised manuscript inclueding all changes and the new Figure 2. Finally, a native English speaker has revised the final version of the manuscript to improve the readability.

Reviewer 2 Report

I have reviewed the manuscript. This is a nicely written review on secondary MDS.  I have one comment/suggestion.

1) I think one part on the management of secondary MDS should be added to the manuscript, since in the "Simple Summary" part, the authors stated that they would also be summarizing the clinical management of this myeloid malignancy.

Author Response

We agree with Reviewer 2 and we have included a brief sentence regarding the particular pathophysiology and clinical management of the MDS-pCT patients (see text in red in pages 2 and 4). Finally, a native English speaker has revised the final version of the manuscript to improve the readability.

Reviewer 3 Report

Calvete et al.’s review describes the current controversy concerning the classification of secondary myelodysplastic neoplasms (MDS). Recent knowledge on the causes and pathophysiology of seconday MDS, inherited or acquired genetic predisposition, and CHIP- or treatment-dependence of certain secondary MDS are well described, clear and up to date. The bibliography is also up to date and exhaustive. The complexity of secondary MDS is well explained, as are the insufficiencies of the present classification, and the necessity for an improved, more detailed classification. Thus, the review by Calvete et al. is interesting and deserves publication in this special Issue of Cancers.

However, prior to publication efforts to improve Figure 2 (and legend) page 8 would be appreciated. The present Figure 2 is entitled « Classification of secondary MDS » but in fact, this figure shows 8 scenarios that may lead to 3 categories of MDS : Secondary MDS with germline predisposition ; CHIP dependant secondary MDS ; MDS-pCT (« Sproradic tumors » and « syndrome » are not clear to this reviewer). To make the proposed new classification more easily understandable, my suggestion is to re-group scenarios that lead to each of the 3 types of secondary MDS (rather than classify them by « independence » or « concomittance »).

Calvete et al.’s review describes the current controversy concerning the classification of secondary myelodysplastic neoplasms (MDS). Recent knowledge on the causes and pathophysiology of seconday MDS, inherited or acquired genetic predisposition, and CHIP- or treatment-dependence of certain secondary MDS are well described, clear and up to date. The bibliography is also up to date and exhaustive. The complexity of secondary MDS is well explained, as are the insufficiencies of the present classification, and the necessity for an improved, more detailed classification. Thus, the review by Calvete et al. is interesting and deserves publication in this special Issue of Cancers.

However, prior to publication efforts to improve Figure 2 (and legend) page 8 would be appreciated. The present Figure 2 is entitled « Classification of secondary MDS » but in fact, this figure shows 8 scenarios that may lead to 3 categories of MDS : Secondary MDS with germline predisposition ; CHIP dependant secondary MDS ; MDS-pCT (« Sproradic tumors » and « syndrome » are not clear to this reviewer). To make the proposed new classification more easily understandable, my suggestion is to re-group scenarios that lead to each of the 3 types of secondary MDS (rather than classify them by « independence » or « concomittance »).

Author Response

We appreciate the comments and suggestions of reviewer 3. We have revised the legend of the Figure 2 to clarify the explanation of the different scenarios (see text in red in Figure 2 Legend). However, the problem of re-group the proposed scenarios is that they are overlapping, which is one of the important highlights of the manuscript. The risk factors are grouped as suggested by the referee only in the current classification (section A). The main aim of this manuscript is to alert from grouping the risk factors because they are overlapping. I.e. Few scenarios included in the genetic predisposition group are also, in fact, scenarios for CHIP dependent secondary MDS. On the other hand, independent and concomitant tags are referring to those scenarios where both tumors do not share any etiology or they are related, respectively. However, we have tried to clarify the Figure by adding a “Risk Factors Involved” heading. Finally, a native English speaker has revised the final version of the manuscript to improve the readability.

Round 2

Reviewer 1 Report

Although I do not consider this manuscript a major contribution to the field, I have to recognize that the authors tried to solve my concerns.